# The Risk Factors and Screening Uptake for Prostate Cancer: A Scoping Review

**DOI:** 10.3390/healthcare11202780

**Published:** 2023-10-20

**Authors:** Seidu Mumuni, Claire O’Donnell, Owen Doody

**Affiliations:** 1Department of Nursing and Midwifery, University of Limerick, V94 T9PX Limerick, Ireland; seidu.mumuni@ul.ie (S.M.); claire.odonnell@ul.ie (C.O.); 2Health Research Institute, Department of Nursing and Midwifery, University of Limerick, V94 T9PX Limerick, Ireland

**Keywords:** prostate cancer, risk factors, screening uptake

## Abstract

Objectives: The purpose of this scoping review was to identify the risk factors and screening uptake for prostate cancer. Design: Scoping review. Methods: Arksey and O’Malley’s framework guided this review; five databases (Cumulative Index to Nursing and Allied Health Literature (CINAHL), MEDLINE, PsycINFO, Academic Search Complete and Cochrane Library) and grey literature were searched. Screening was undertaken against predetermined inclusion criteria for articles published before July 2023 and written in English. This review is reported in line with PRISMA-Sc. Results: 10,899 database results were identified; 3676 papers were removed as duplicates and 7115 papers were excluded at title and abstract review. A total of 108 papers were full-text reviewed and 67 were included in the review. Grey literature searching yielded no results. Age, family history/genetics, hormones, race/ethnicity, exposure to hazards, geographical location and diet were identified as risk factors. Prostatic antigen test (PSA), digital rectal examination (DRE), transrectal ultrasound (TRUS), magnetic resonance imaging (MRI), magnetic resonance spectroscopic imaging (MRSI) and prostate biopsy were identified as screening/diagnostic methods. The evidence reviewed highlights moderate knowledge and screening uptake of prostate cancer with less than half of men reporting for PSA screening. On the other hand, there is a year-to-year increase in PSA and DRE screening, but factors such as poverty, religion, culture, communication barriers, language and costs affect men’s uptake of prostate cancer screening. Conclusion: As prostate cancer rates increase globally, there is a need for greater uptake of prostate cancer screening and improved health literacy among men and health workers. There is a need to develop a comprehensive prostate cancer awareness and screening programme that targets men and addresses uptake issues so as to provide safe, quality care. Strengths and limitations of this study: (1) A broad search strategy was utilised incorporating both databases and grey literature. (2) The PRISMA reporting guidelines were utilised. (3) Only English language papers were included, and this may have resulted in relevant articles being omitted.

## 1. Background

Prostate cancer is the second-most prevalent cancer in men and was the fifth-most common cause of cancer-associated death among men in 2020 [1,2]. Peripheral zone cancer accounts for 70% of cases, with transition and central zone cancers accounting for 20% and 10%, respectively [3]. Cancer cells may travel through the blood and lymphatic fluid, causing metastasis mostly commonly affecting bone, followed by lymph nodes, lung and liver [4]. Adenocarcinomas contribute 95% of all prostate cancers, whereas the occurrence of primary carcinoid, sarcomas and primary small cell carcinomas are rare, and thus it is essential to identify persons at risk and highlight the importance of screening uptake [5]. Prostate cancer is a disease of global importance, with about 1,414,000 new cancer cases diagnosed and over 375,304 deaths in 2020 [6]. While the incidence of prostate cancer can vary across geographical regions [7,8], it is recognised by the World Health Organization as the third most commonly diagnosed cancer worldwide, with 1,414,259 cases [9]. Oceania and North America report the highest incidence rates, 79.1 and 73.7 per 100,000 men, respectively, followed by Europe (62.1), Africa (26.6) and Asia (11.5) [2]. The incidence of prostate cancer is found to increase with age from one in 350 men under the age of 50 to one in every 52 men aged 50 to 59, and nearly a 60% increase is seen in men over 65 years of age [1,10]. Arguably, while some report that the high incidence may be due to over-diagnosis of prostate cancer and voice caution regarding the benefit of protein-specific antigen screening (PSA) [11], the mortality rate for prostate cancer still varies proportionally with increase in age, with 55% of prostate cancer deaths occurring in men over 65 [2].

Frequently described symptoms of prostate cancer include urinary disorders which affect frequency and ease of urination [12], weak urine flow, fullness of bladder, poor urine urge control, painful urination and nocturia [13,14], urinary retention and back pain [15]. Although many symptoms exist, unfortunately, a key aspect of prostate cancer is its indolent course, which in its curable stages is asymptomatic [2,16]. Often men with co-morbidities with prostate cancer are more likely to die from the existing co-morbidity rather than prostate cancer [9], resulting in some men being diagnosed with prostate cancer during autopsy [7,17]. This highlights the need for greater awareness of prostate cancer and identification of the risk factors and issues with screening utilisation. The specific cause of prostate cancer is unknown; however, men who fall within specific risk factor groups are more likely to develop prostate cancer [1].

These groups depend on the presence of both modifiable (alcohol consumption, obesity, smoking, diet high in saturated fats) and non-modifiable risk factors (advancing age, race, family history) and may significantly increase the risk of developing prostate cancer [16]. In addition, the intake of a high-fat diet [18] and obesity [19] increase the risk, and there is almost double or triple the risk of developing prostate cancer among black, Caribbean and African American men [20]. Men with a family history of prostate cancer are relatively more at risk of being diagnosed with prostate cancer [21,22], and the risk increases with closeness of the familial link and the total number of family members affected [23].

Based on prevalence, environmental and generic risk factors, screening is emphasized for men [21]. Protein-specific antigen (PSA) screening has been used to confirm prostate cancer presence based on elevated levels of glycoprotein, where values greater than 4.0 ng/mL indicate high risk [24]. However, Rawla [2] argues that protein-specific antigen levels are raised in some men without cancers, and similarly, Heidegger [22] indicates that urinary tract infections, drugs or surgery can alter the levels of protein-specific antigen in the body. Thereby, the prime standard to confirm prostate cancer is a tissue biopsy, as the PSA screening tests appear to be clinically insignificant in the diagnosis of prostate cancer [24], and different guideline standards for PSA screening have resulted in over-testing and unnecessary biopsies have raised concerns in healthcare practice [5]. Furthermore, a digital rectal examination may be conducted to observe enlargement and abnormalities of the prostate gland [3] and indicate the necessity for further investigation (prostate biopsy) in the diagnosis of prostate cancer. There is support for providing routine PSA screening to individuals whose risk factors indicate the need for screening and are between the ages of 55 and 69 [25]. The European Association of Urology (EAU) guidelines do not recommend widespread mass screening for prostate cancer but do strongly recommend early detection in men and a need for positive health literacy [26]. However, in case of further investigation, a prostate biopsy is an important element in the diagnosing process [24]. Given the need for greater awareness of prostate cancer, its risk factors and diagnosis, this paper aims to identify and scope the risk factors and screening uptake for prostate cancer in men.

## 2. Methodology

A scoping review was conducted to help map the landscape of the existing literature to ascertain relevant articles [27]. Arksey and O’Malley’s scoping review approach [28] and Levac et al.’s methodological update [29] were used to identify relevant studies for the chosen topic, as suggested in accordance with conducting a scoping review. This framework consists of five steps: (1) identifying the research question; (2) identifying relevant studies; (3) selection of studies; (4) charting the data; and (5) collating, summarising, and reporting results. To conduct and report this review, the Preferred Reporting Items for Systematic Reviews and Meta-Analyses Scoping Review—PRISMA-ScR [30] (Appendix A) was utilised along with a PRISMA flow diagram [31] (Figure 1), PRISMA-S (search) [32] (Appendix A) and PRISMA-Ab (abstract) [31] (Appendix A).

### 2.1. Identifying the Research Question

The PEO method (Table 1) was used to identify relevant elements of the research question of “What are the risk factors and screening uptake of patients with prostate cancer?” To answer the review question, the following objectives were identified:What risk factors for prostate cancer are evident within the literature?What screening and diagnostic tools for prostate cancer are evident within the literature?What is the screening uptake for prostate cancer evident within the literature?

### 2.2. Identifying Relevant Studies

#### Search Strategy

To guide the development of the search strategy and key terms, a preliminary exploration was undertaken to help provide understanding and guidance for the search terms and direction of the search. To capture the wide-ranging nature of this review, a wide range of keywords/search terms were developed for use within the search. These keywords/search terms assisted in developing the search strategy (Table 2), and the search was performed within five databases: Cumulative Index to Nursing and Allied Health Literature (CINAHL), MEDLINE, PsycINFO, Academic Search Complete, Cochrane Library and grey literature. The search was initially conducted in 2022 to develop and agree upon the search strings, and the final search was updated on 30 June 2023; database search outputs are available (https://doi.org/10.6084/m9.figshare.23686167, accessed on 14 July 2023). The search also combined terms from MESH/Subject heading/Thesaurus. The search was performed in (AB) Abstract/Author-Supplied Abstract and (TI) Title search functions. The search process utilised Boolean operators (‘OR’/‘AND’) to search for records containing either search terms or combine search strings. All records were transferred to Endnote Library 2021 (Clarivate Analytics, Philadelphia, PA, USA) where duplicates were identified and removed. To assist the screening and voting process, the remaining records were then transferred to Rayyan (Qatar Computing Research Institute). The grey literature search included the following sources: the CADTH ‘Grey Matters’, Open Grey, the National Institute for Health and Care Excellence (NICE) and the World Health Organization International Clinical Trials Registry Platform (WHO ICTRP); search outputs are available (https://doi.org/10.6084/m9.figshare.23686185, accessed on 14 July 2023).

### 2.3. Selection of Studies

The screening processes were conducted within Rayyan (Qatar Computing Research Institute) by two reviewers (SM, OD) utilising inclusion criteria (Table 3). Titles and abstracts of all records were screened against the inclusion criteria and the full texts of all records that may meet the inclusion criteria were gathered and examined and those that met the inclusion criteria went forward for data extraction and inclusion in the review.

### 2.4. Charting the Data

Data extraction is one of the most important activities within the review process, and to assist this process the authors used a data extraction table (Appendix A) to extract and record all data from the included papers. This process was conducted by reading each paper, extracting related information to the review question and entering it into the data extraction table. This scoping review mapped the existing literature in terms of nature, characteristics and source of evidence, extracting summaries from each paper in the data extraction table. The data extraction table utilised the author/s, year, title, country, aim or focus, methods and methodologies, summary of findings, recommendation made and limitations. The use of the data extraction table ensured that all relevant data were consistently extracted from each paper and assisted with mapping and coding the data.

### 2.5. Collating, Summarising and Reporting Results

Arksey and O’Malley’s final stage involves summarising and communicating findings. Within this stage, 67 papers across 23 countries were identified and included in this review. All 67 papers were read, and summary data were extracted to communicate the study characteristics and address the objectives of the review. This process enabled the data to be mapped, charted and reported clearly within each of the three objectives.

### 2.6. Patient and Public Involvement

While no formal patient and public involvement arrangement was developed for this review, two persons with prostate cancer (one survivor and one active) contributed to this paper by reading the findings and sharing their experience to assist the authors in understanding and placing the findings and discussion in context.

## 3. Results

The search generated 10,899 papers across the five databases, and after the removal of duplicates (*n* = 3676), 7223 papers remained for screening. These papers were screened at the title and abstract stages, resulting in 7115 papers being excluded. The remaining 108 papers went forward for full-text review. The full-text review identified 67 papers that met the inclusion criteria, and the reasons for exclusion (*n* = 41) are reported in the PRISMA flow diagram (Figure 1).

The grey literature search revealed 10,421 results, and all but 18 were removed at the title and abstract stages. The 18 grey literature full texts were screened and all were excluded, resulting in none meeting the review criteria and being eligible. During the process, the authors met to discuss papers and to reach agreement if they were in any doubt about a particular paper. This review is reported in line with the Preferred Reporting Items for Systematic Reviews and Meta-analysis for Scoping Reviews (PRISMA-ScR) [30], PRISMA search [32] and PRISMA abstract [31].

The retrieved literatures were qualitative (*n* = 3), quantitative (*n* = 27), discussion/expert opinion papers (*n*= 33), systematic reviews (*n* = 2), editorials (*n* = 1) and mixed method (*n* = 1). In terms of countries of publication, *n* = 33 were from the United States of America; *n* = 4 from both the United Kingdom and Canada; *n* = 3 from Sweden; *n* = 2 each from Brazil, Jamaica, Japan and Burkina Faso; and *n* = 1 from each of the remaining countries, Australia, Ghana, Malaysia, China, Mexico, Cameroon, South Korea, Nigeria, United Arab Emirate, Portugal, Taiwan, Netherlands, Finland, South Africa and Switzerland. Both the qualitative and quantitative sample sizes ranged from 25 to 351,448 participants. Questionnaires were mostly used as data collection methods, of which SPSS, Chi-squared and deceptive analysis were used to analyse the study data. Almost all the qualitative, quantitative and mixed method studies obtained ethical approval before conducting the study.

(A)What risk factors for prostate cancer are evident within the literature?

Within the review, eight risk factors for prostate cancer were evident, with family history/genetics evident in 33 papers [33,34,35,36,37,38,39,40,41,42,43,44,45,46,47,48,49,50,51,52,53,54,55,56,57,58,59,60,61,62,63,64,65], age in 31 papers [34,35,36,37,38,39,40,43,44,45,46,47,48,49,50,51,52,53,54,55,57,58,60,61,63,64,65,66,67,68,69], race/ethnicity in 29 papers [34,35,36,38,39,40,43,45,46,47,48,50,52,53,54,55,57,58,60,61,62,63,64,65,66,68,70,71,72], diet/weight in 17 papers [34,36,37,38,40,43,45,47,51,52,53,57,60,63,64,67,73], occupation/environmental/lifestyle in 15 papers [33,34,36,41,43,44,45,46,54,62,63,64,67,74,75], geographical location in 8 papers [36,42,44,46,60,63,64,76], hormone/infertility/vasectomy in 5 papers [37,40,41,52,62] and exposure to hazards in 4 papers [54,61,62,74]. Family history/genetics was widely accepted and an undisputed risk factor, with men with a family member with a history of breast cancer or those with gene (HOXB13) identified as more likely to develop prostate cancer [46,47]. Age as a risk for prostate cancer was seen to begin at age 40 [47,62,77,78], and into the fifth, sixth, and seventh decades of life men become more prone to prostate cancer [33,47,49,51,62,72,77,79]. However, Chang [41] identified no significant risk with age and prostate cancer. In terms of race and ethnicity, African American [34,36,55,62,68,71,72,80,81], Caribbean [36,66] and black men [35,36,57,70,77,82] were more likely to develop prostate cancer, while Adler et al. [74] identified no significant difference in ethnicity. Diet and the intake of red meat [44,45,53], high saturated fats [39,40,43,45,51,63], sugars (sweets and beverages) [58] and a high sodium diet were identified as risks of developing prostate cancer [45]. However, diet was seen as a weak association [37], and weight had no significant difference [41] in terms of prostate cancer risk.

Occupation, environment and lifestyle were factors with risks relating to sedentary lifestyle [64,74], shift work or management role [74], smoking and alcohol [35,36,37,62,63,64,74,75] and the number of lifetime partners [41]. However, the work of Childre et al. [43] indicates that there is no effect from the number of lifetime partners, and Bergengren et al. [37] highlight that the level of physical activity has no clear relationship with increased risk of prostate cancer. Hormones [40,52,60], infertility [37] and vasectomies [37,41,43,52,62] were identified as risk factors, and the presence of high levels of hormones (male testosterone) [40] may occur due to the excess production hormones activating carcinogens [34]. However, the relationship between hormones and vasectomies and prostate cancer was questioned, with Sasagawa and Nakada [60] finding no evidence of a correlation with hormones and Childre et al. [43] finding no evidence of a correlation with vasectomies. Geographical location (e.g., Vietnam) can serve as a predisposing factor for prostate cancer [46,63,64] and may be due to access issues [51,56,66,81,82,83,84]. Finally, men exposed to hazards (e.g., benzene, toluene, pesticides, fuels, solvents, radiation, agent orange) were identified as being at an increased risk of prostate cancer [34,54,62,74], and generally military roles were highlighted [54,74].

(B)What screening and diagnostic tools for prostate cancer are evident within the literature?

Within this review, the screening and diagnostic process evidence included the following: prostatic antigen test/biomarkers were present in 50 papers [33,35,36,37,38,39,40,42,43,44,47,48,49,50,51,52,54,55,56,57,58,59,61,63,64,65,66,68,69,70,71,72,76,77,78,79,81,82,83,85,86,87,88,89,90,91,92,93,94,95], digital rectal examinations were present in 27 papers [36,38,39,40,43,47,48,50,51,52,57,61,63,64,65,66,68,71,72,78,79,82,89,91,92,94,95], transrectal ultrasounds were present in 6 papers [47,48,52,69,72,95], endorectal coil magnetic resonance imaging and magnetic resonance spectroscopic imaging were present in 2 papers [36,95] and prostate biopsies were present in 19 papers [36,38,39,40,47,48,50,52,55,61,63,64,65,68,69,72,89,94,95].

The prostatic-antigen test (PSA) was the most-identified screening tool which involves the use of a blood sample to measure the level of PSA in the body, which helps in the diagnosis of the condition. However, PSA testing has raised some controversy [85] and concerns regarding its limitations [36], the threshold level [77,88] and overdiagnosis and overtreatment [38,59,64,85,87,88,89]. Digital rectal examination (DRE) was accepted as a primary tool used by healthcare professionals in screening for prostate cancer, and other screening tools identified were the use of tissue samples to detect any abnormality that might cause prostate cancer, transrectal ultrasound (TRUS) to gain an image of the prostate gland, endorectal coil magnetic resonance imaging (MRI) and magnetic resonance spectroscopic imaging (MRSI). Digital rectal examination and prostatic antigen tests were seen as the first line of screening and as necessary before a biopsy is undertaken to confirm diagnosis [47,52,63,65,72,94,95].

(C)What is the screening uptake for prostate cancer evident within the literature?

With regard to screening uptake, the majority of participants (83.9%) recognise the need for men to attend prostate cancer screening even without any signs or symptoms [71]. Half of Canadian men aged 50 (47.5%) reported a lifetime PSA screening event [35], while Kabore [92] indicated that most participants (70.3%, *n* = 422) were unaware of any screening tests for prostate cancer being available. There is evidence of screening with PSA and DRE increasing year-by-year [82]. However, although clinicians recommend prostate cancer screening for their patients, 10.28% would not consider undergoing a PSA test themselves [49], and within the studies reviewed, prostate cancer screening knowledge was moderate among the majority of men [56,82,84,92,96]. The evidence points to racism and poverty; cultural, religious, language and communication barriers; high medical expenses; and geographical locations and knowledge/awareness as factors that hinder men from assessing prostate cancer screening [35,56,71,80,82,83,84,90,92,96,97,98]. Within screening uptake, the decision-making process was a key factor, with patients (men) not being included [49,85] or not informed [81] in the process. This highlights the need to include patents in an informed decision process [34,87,91]; in terms of the use of decision aids [86] and where decision-making was a shared decision, it related to men from or in the military [54]. Other factors that affected low screening uptake were education level [83,93,97,98,99], income or socioeconomic class [35,80,81,93,97], being a black man [36,57], being American Indian [93], being African American [80,81], culture or religion [96], knowledge, awareness and beliefs [56,84,92] and fear or distrust [51,90]. Positive predictors of screening uptake were smokers on night duty shift work [35], those married [93,99] and those with prostate knowledge [98].

## 4. Discussion

Although a definitive cause for prostate cancer may be difficult to specify, many aspects were evident within this review. Family history and genetics were identified in 33 papers as a risk for developing prostate cancer. Having a direct family member with prostate cancer can double the chance of an individual developing prostate cancer [100,101], and if two or more direct family members have been diagnosed with prostate cancer, it quadruples one’s chance of developing prostate cancer [102,103]. Furthermore, having a family member with breast cancer can increase the chance of developing prostate cancer due to their genetic makeup (e.g., BRCA1, BRCA2, HOXB13, CHEK2, HOXB13 and ATM) [13,14]. Age was identified as a major predisposing factor in 31 papers, and within the wider literature prostate cancer risk varies consistently with age, with incidence rates increasing rapidly for each successive period of life [104,105]. Globally, age is strongly linked to the development of prostate cancer; older men are more likely to be diagnosed with prostate cancer and have a lower rate of survival [106]. While prostate cancer can occur before the age of 45, which has been related to stress, alcohol consumption and smoking [100,107], most cases are detected and diagnosed after 50 to 55 years [2]. Race/ethnicity was a risk factor for prostate cancer in 28 papers; specifically mentioned were African American men, Caribbean men and black men. The wider literature highlights that prostate cancer is mostly detected and diagnosed in African Americans [108,109], Caribbean men [110,111] and black men [110,112] more than Caucasian men [113] and Asian men [114] due to their diet and lifestyle [22].

However, Pagadala et al. [101] state that the reason is not known, and some studies correlate prostate cancer with genetics, diet, hormones and environmental factors. Hormones are seen as a predisposing factor for prostate cancer, with androgen being seen as responsible for the growth of prostate cells [103,115] and high testosterone levels. This can be seen as causing cell stimulation, thereby increasing prostate cell activity and leading to the onset of prostate cancer [116]. While dietary intake is identified as a risk factor, the role diet plays in causing prostate cancer is unknown [117]. However, high intake of fatty foods, calcium or sodium [2,14] and excess consumption of coffee, sugar-sweetened beverages and dairy products like milk and cheese can predispose one to developing prostate cancer [118]. Arguably, the intake of other food products such as soy, which contains protein and phytoestrogens, lycopene and food rich in vitamin E may help to reduce prostate cancer [105]. Papers linked prostate cancer development to chemical agents such as agent orange used in the Vietnam war, and chemicals such as benzene, toluene, xylene and styrene found in some occupations such as military and fire service may lead to men developing prostate cancer [119]. Of interest was the fact that smokers had a higher screening rate [35]; however, this was within a population who are also night duty workers, and it is difficult to truly state which of these two factors led to higher screening uptake. Geographical areas such as North America, northwestern Europe, Australia and the Caribbean Islands show higher incidences of men developing prostate cancer [2], whereas geographical locations like Asia, Central America and South America have recorded low incidences of prostate cancer, and this may be due to different lifestyles and eating habits, exposure to radiation and prostate cancer screening [117].

Screening is significant in the early detection and diagnosis of asymptomatic prostate cancer, which will help prolong life [120]. To conduct accurate, reliable and easy-to-administer tests, this review identifies digital rectal examination (DRE) and the prostatic antigen test (PSA) as the tools most utilised for early detection of the condition [121]. PSA was the most common screening tool identified in this review and involves a simple blood test to measure the level of PSA in the body [13,14]. However, several controversies have been raised surrounding the potency or accuracy of the PSA test [122]; despite this, the use of PSA is still recommended in many countries [4,123]. Such controversy relates to issues of over-screening [124], overdiagnosis [125] and the effect of overdiagnosis on the individual [126]. DRE is the screening test primarily chosen by clinicians to detect prostate cancer [117].

The choice of tool used needs to be considered in light of its pros and cons and what is suitable to each individual. PSA is traditionally recommended early in the process but varies internationally due to its issues with overdiagnosis [38,59,64,85,87,88,89]. Most prostate cancer detection with DRE occurs in the advanced stage of the condition, thereby impeding treatment options and quality of life [17,127]. Prostate biopsy is a tool for detecting and diagnosing prostate cancer which investigates tissue samples from the prostate gland under a microscope to detect any abnormality or carcinoma [24]. However, a prostate biopsy is used at the end of the screening and diagnosis process or when DRE and PSA fail to give conclusive results [2]. Less frequently used screening tools are a transrectal ultrasound (TRUS), endorectal coil magnetic resonance imaging (MRI), magnetic resonance spectroscopic imaging (MRSI) and biomarker tests like SelectMDx, but they are identified as screening tools used to increase the possibility of providing accuracy in the early detection and diagnosis of prostate cancer [128,129]. However, TRUS biopsy has risks such as pain, haematuria, acute urinary retention, haematospermia, infection, rectal bleeding, sepsis and erectile dysfunction [130]. Evidence supports the use of MRI for men at risk of harbouring prostate cancer and who have not undergone a previous biopsy and in men with an increased prostate specific antigen following an initial negative standard prostate biopsy procedure [131]. MRSI is a non-invasive method based on spectroscopic analysis of tissue metabolism, which aids clinicians in identifying cellular biochemistry by detecting high levels of choline compounds and providing details of tumour volume and metabolite behaviour, which can be critical in deciding care options [132].

Despite screening being an important tool in the early detection of prostate cancer, the findings in the review highlight that up to 70.3% of respondents can be unaware of the availability of any screening tests for prostate cancer [92], and less than half at age 50 experienced a lifetime PSA screening event [35]. Potential obstacles to cancer screening utilisation among men include language barriers, health literacy, misconceptions regarding cancer risks and lack of knowledge regarding access to cancer screening services [133,134]. Early detection screening programmes should focus on cancers causing morbidity and/or death, as shown by reduced cancer-related mortality [135]. The findings of the review indicate that knowledge of prostate cancer screening is moderate among the majority of men, and a 2018 study in Kenya highlights an uptake rate of prostate cancer screening as low as 1.3% [136]. These rates highlight that despite public consultation/promotion regarding prostate cancer and screening, uptake is still low and issues like race, poverty, culture, religion, language/communication barriers, cost, and geographical location are recurring factors hindering prostate cancer screening uptake [128,129]. Any awareness or uptake programme that fails to address such issues will have difficulty in addressing men’s health and the provision of quality care.

Of concern was that although clinicians (general practitioners) recommend prostate cancer screening to their patients, 10.28% would not consider undergoing a PSA test themselves [49]. This may indicate an unconscious bias among some clinicians which may affect the information provided and the decision-making process. Therefore, an open discussion with full disclosure is essential in the decision-making process and in the provision of quality service [137,138]. However, it must also be recognised that is some cases even where clinicians have involved the patient in the decision-making process, it has been noted that African American men can forgo screening due to previous experiences or stigmatisation [139]. A driving force for prostate screening was marriage and women suggesting to their husbands to regularly undergo prostate screening even without any signs of any carcinoma [140]. However, on the other hand, while partners within the LGBT community agree screening helps early detection and treatment, there is a low uptake rate [141]. While this may indicate a bias or stigma, it must be considered in terms of the underrepresentation of the LGBT community in cancer research; however, a recent study by Ma et al. [142] indicated that gay/bisexual cohorts were more likely to participate in prostate screening and more likely to make informed decisions. Therefore, to generally increase the overall screening uptake for prostate cancer, there needs to be education of the healthy population that are not patients, targeted health promotion campaigns that address knowledge and decision-making and engagement of policy makers [143].

## 5. Limitations

There are a number of strengths and limitations to this review. Firstly, only one author (SM) conducted the data extraction and coding. However, to offset the risk of bias and errors in the process, a second author met to discuss each stage of the process, e.g., screening, inclusion criteria, coding and extraction, and a second author (OD) verified 20% of the dataset. While a protocol was in place to guide the project and frequent support and supervision were provided by the other authors, the protocol was not published prior to the review being conducted. While this review used precise, transparent methods based on study and reporting guidelines [28,30], a quality appraisal and risk of bias assessment were not conducted, as the focus of this review was on identifying, mapping and charting the risk factors and screening uptake for prostate cancer. Thus, this paper may only offer a descriptive account of available data. Furthermore, there was no formal patient and public involvement and there was an opportunity for engagement, following recent published guidance on stakeholder involvement in systematic reviews [144]. Data in this review included primary and secondary data, and the inclusion of secondary data can be seen as both a strength and limitation. In addition, Whittemore and Knafl highlight that database searching has limitations such as inconsistencies in search terminology and indexing problems, which may yield only up to 50% of eligible studies [145]. Despite these limitations, the authors acknowledge that this is the first scoping review based on a comprehensive literature search to identify the current state of knowledge regarding risk factors and screening uptake for prostate cancer, providing an overview of the available evidence and a focus for healthcare professionals, health promotion practitioners and service providers in an area where evidence, knowledge or awareness is lacking.

## 6. Conclusions

As prostate cancer rates are increasing, prostate cancer screening uptake needs to be encouraged and improved by targeted education for men and health workers that addresses a comprehensive prostate cancer awareness programme to assist in improving health literacy. This needs to be supported by policy makers so that screening uptake and quality care provision are improved. This review highlights issues relating to prostate cancer knowledge, risks, screening and screening uptake, and there is a need to understand barriers and enablers to screening uptake through identifying people’s perspectives and experiences. While studies have linked prostate cancer to various factors, i.e., age, race, ethnicity, family history, genetics, hormones, lifestyle, diet, location, environmental and occupational exposures, exposure to hazards, hormones, vasectomies and infertility, more evidence is required on the risk factors and preventive methods of prostate cancer development, and it is important for healthcare practitioners to utilise such information to educate patients and encourage prevention. While barriers exist for some people in accessing prostate cancer screening, this review highlights a lack of knowledge, which results in ignorance, misconceptions and mistrust. Underpinning this lack of knowledge is education, and without knowledge it is difficult to comprehend information on prostate cancer screening. However, in light of the fact that prostate cancer incidence is not equally distributed and accessibility of screening, diagnosis and treatment differs across countries, it is important for countries to identify their specific issues and needs and respond appropriately. Generally, governments should prioritise prostate care along with other curative and preventative care approaches, and recommendations include (a) governments developing policies and programmes for prostate care and screening, (b) educating the public about prostate cancer, (c) future research to determine the effect of educational interventions and (d) government support for prostate cancer screening.

## Figures and Tables

**Figure 1 healthcare-11-02780-f001:**
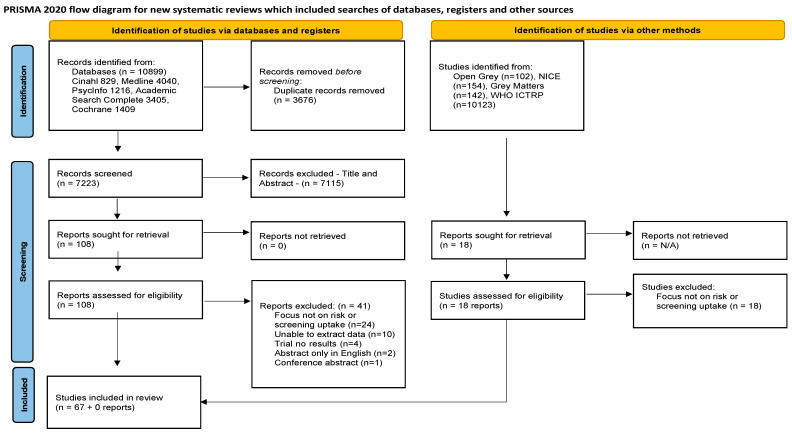
Characteristics [31].

**Table 1 healthcare-11-02780-t001:** PEO framework.

P	Population	Males
E	Exposure	Prostate cancer
O	Outcome	Risk factors and/or screening uptake

**Table 2 healthcare-11-02780-t002:** Search terms.

Search	Search Terms
S1	(MM “Prostate”) OR (MM “Prostatic Neoplasms+”)
S2	(MM “Neoplasms, Second Primary”)
S3	prostate cancer OR prostatic neoplasm * OR prostate carcinoma (Title or Abstract)
S4	(MM “Early Detection of Cancer”)
S5	(MM “Mass Screening+”)
S6	screening OR assessment * OR test * OR diagnosis OR early detection OR detect * (Title or Abstract)
S7	risk factor * OR contributing factor * OR predisposing factor * (Title or Abstract)
S8	S1 OR S2 OR S3
S9	S4 OR S5 OR S6
S10	S7 AND S8 AND S9

**Table 3 healthcare-11-02780-t003:** Inclusion and exclusion criteria.

Inclusion Criteria	Exclusion Criteria
Grey literature and peer reviews or primary research articles.	Treatment and management of prostate cancer.
Articles addressing risk factors and screening for prostate cancer.	Articles published after 30 June 2023.
Articles published before 30 June 2023.	Non-English papers.

## Data Availability

No data are available other than what is reported in this review and what is available in the original published papers used in this review.

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
