# Peer review of "The Risk Factors and Screening Uptake for Prostate Cancer: A Scoping Review"

_healthcare, 2023, doi:10.3390/healthcare11202780_

Round 1

Reviewer 1 Report

First of all, I want to congratulate the authors for their manuscript “The risk Factors and Screening Uptake for Prostate Cancer: A Scoping Review”. 

In their review, the authors present 67 out of 10899 papers, which fulfilled the inclusion criteria for their scoping review. They identified 8 risk factors for prostate cancer (family history/genetics evident, age, race/ethnicity, diet/weight, occupation/environmental/ lifestyle, geographical location, hormone/infertility/vasectomy, exposure to hazards), 6 screening tools (PSA, DRE, prostate biopsy, transrectal ultrasound, endorectal coil magnetic imaging, magnetic resonance spectroscopic imaging) and presented the screening uptake for prostate cancer. 

Since prostate cancer is widely spread and has a high impact on patients´ life and public health care systems, it is important to endorse studies/reviews dealing with risk-factors and screening for publication. 

But in my opinion, some points have to be revised before acceptance of the manuscript: 

-       All mentioned screening tools, despite prostate biopsy itself, help urologists to make the indication for biopsy. The only way to diagnose prostate carcinoma is via biopsy. So, it is not correct to list prostate biopsy as a screening tool, alongside with PSA, DRE, etc. 

-       On the other hand, PSA, DRE, ultrasound or magnetic resonance imaging do not diagnose prostate cancer. They can give urologists a hint, whether it is more or less likely to diagnose prostate cancer by biopsy. Again, this paragraph should be reformulated for a better understanding. 

-       E.g.: Line 338-339: prostate biopsy is NOT seen to be used as the last resort after failed DRE and PSA! Prostate biopsy always stands at the end of screening to diagnose PCa. 

-       The geographical origins of the papers should have an influence in the analyzes. It is very likely that the knowledge of prostate cancer and therefore the screening uptake differ between countries. To state, that “knowledge of prostate cancer screening is moderate among majority of men” (line 265/266) is inappropriate when viewed globally. 

-       Do the analyzed papers provide explanations about the dependencies for “prostate knowledge”? I believe that this is important, since prostate knowledge is a positive predictor for screening uptake. 

-       Is there an explanation, why “Smoking” leads to a higher screening uptake? Interesting issue to discuss!

-       Line 296-297: It is not permitted to state, that prostate cancer diagnosed after 50 to 5 years occur because of stress, alcohol consumption and smoking! Rephrasing obligate.

-       Line 329-332: PSA is still recommended. This is very depending which guideline you cite and not applicable globally! Rephrasing obligate. 

-       Line 305-307: Difficult to understand. It is evident, that testosterone is obligate for prostate cancer development. 

-       The authors postulate several times, that physicians recommend prostate screening to their patients but only a small percentage would consider undergoing screening themselves. Which specialization do the physicians have? Are they urologists? I believe these physicians are other specialists and therefore this could be another hint for “lack of prostate knowledge”. This issue has to be discussed more precisely.  

-       Overall, the “Discussion” should be more a discussion. Especially at the beginning results are continuously presented without discussing them. 

-       Conclusion: In my opinion it is not applicable to conclude that this review - like it is presented and discussed so far – highlights “the lack of knowledge, which results in ignorance, misconceptions and mistrust”. 

Last section of the conclusion: the situation about knowledge of prostate cancer, accessibility for screening, diagnostic and treatment differs enormously among the globe. In addition to that prostate cancer incidence is not equally distributed.   Therefore, it is important to present where the data is coming from and recommendations should relate to these areas. 

Thank you very much for giving me the chance to read and review this manuscript.

Author Response

Reviewer 1

Answer

All mentioned screening tools, despite prostate biopsy itself, help urologists to make the indication for biopsy. The only way to diagnose prostate carcinoma is via biopsy. So, it is not correct to list prostate biopsy as a screening tool, alongside with PSA, DRE, etc. 

Thank for your comments, we have added screening and diagnostic tool to the heading to provide greater clarity. we appreciate that the PSA and DRE are screening tool used prior to biopsy. After presenting the PSA and the DRE, we refer to other screening tool that specifically use tissue samples to detect any abnormality.

On the other hand, PSA, DRE, ultrasound or magnetic resonance imaging do not diagnose prostate cancer. They can give urologists a hint, whether it is more or less likely to diagnose prostate cancer by biopsy. Again, this paragraph should be reformulated for a better understanding. 

We have rephrased the statement at the start of the section to “the screening and diagnostic process evident included”.

-       E.g.: Line 338-339: prostate biopsy is NOT seen to be used as the last resort after failed DRE and PSA! Prostate biopsy always stands at the end of screening to diagnose PCa. 

Thank you for your observation and comment, this was a mis phrasing and edited to - prostate biopsy is used at the end of the screening and diagnosis process or when DRE and PSA fail to give conclusive results

The geographical origins of the papers should have an influence in the analyzes. It is very likely that the knowledge of prostate cancer and therefore the screening uptake differ between countries. To state, that “knowledge of prostate cancer screening is moderate among majority of men” (line 265/266) is inappropriate when viewed globally. 

Sentenced edited to clarify moderate “within the studies reviewed”

  Do the analyzed papers provide explanations about the dependencies for “prostate knowledge”? I believe that this is important, since prostate knowledge is a positive predictor for screening uptake. 

We have now acknowledged knowledge as one of the factor influencing screening uptake and referenced appropriately.

-       Is there an explanation, why “Smoking” leads to a higher screening uptake? Interesting issue to discuss!

We have added to the sentence - of interest was the fact that smokers had a higher screening rate [35] however, this was within a population who are also night duty workers and it is difficult to truly state which of these two factors led to higher screening.

-       Line 296-297: It is not permitted to state, that prostate cancer diagnosed after 50 to 5 years occur because of stress, alcohol consumption and smoking! Rephrasing obligate.

Thank you we have rearrange the sentence for clarity.

-       Line 329-332: PSA is still recommended. This is very depending which guideline you cite and not applicable globally! Rephrasing obligate. 

We have rephrased to acknowledge “the use of PSA is still recommended in many countries”.

-       Line 305-307: Difficult to understand. It is evident, that testosterone is obligate for prostate cancer development. 

We have edited the sentence and split to form two sentences for greater clarity.

-       The authors postulate several times, that physicians recommend prostate screening to their patients but only a small percentage would consider undergoing screening themselves. Which specialization do the physicians have? Are they urologists? I believe these physicians are other specialists and therefore this could be another hint for “lack of prostate knowledge”. This issue has to be discussed more precisely.  

We have edited physician to “clinicians (general practitioners)” for clarity.

Overall, the “Discussion” should be more a discussion. Especially at the beginning results are continuously presented without discussing them. 

We have edited the discussion for great flow and discussion of points.

Last section of the conclusion: the situation about knowledge of prostate cancer, accessibility for screening, diagnostic and treatment differs enormously among the globe. In addition to that prostate cancer incidence is not equally distributed.   Therefore, it is important to present where the data is coming from and recommendations should relate to these areas. 

Conclusion edited and now acknowledges international differences among countries.

Reviewer 2 Report

Dear Authors,

this article is interesting and actual

The Authors did a scoping review having as objective to identify the risk factors and screening uptake for prostate cancer.  It shows the current state of knowledge about screening worldwide. The authors concluded with some engaging recommendations for governments. 

enjoyed reviewing this manuscript and have only some minor requests for revision: 

Exclude from the key concepts “palliative care” as it seems to be inappropriate for the topic; The subtitles in the “Results” part – A, B, C should be pointed out to improve the readability.

None

Author Response

Reviewer 2

Response

Exclude from the key concepts “palliative care” as it seems to be inappropriate for the topic; - The subtitles in the “Results” part – A, B, C should be pointed out to improve the readability.

We have excluded palliative care from the key concept as suggested.

A, B, C has been edited to improve the readability and visibility as recommended.

Reviewer 3 Report

-          Abstract:

What is the meaning of: “less of half of men reported for PSA screening”?

An important factor that should be added is health literacy.

“The uptake of prostate cancer screening needs to improve”: this is not what is the first aim of the European Commission today: they want to have organized screening programs, that indeed will be successful only when the health literacy among the male population has been upscaled.

The authors also mentioned that “there is the need to develop a comprehensive prostate cancer awareness program to assist in improving screening uptake and providing safe quality care”.  From experience in other countries awareness campaigns do not change very much to the low screening uptake, for instance in Sweden, the organized PSA-testing with an informed invitation letter for PSA testing resulted in no more than 40-45% of men finally being tested. This is pretty low compared to for instance the uptake in breast cancer screening.

-          Background:

The authors mention that “prostate cancer cells cause metastasis most commonly in bones, kidney, ureter and bladder”. For me, bone is the most affected, followed by lymph nodes, lung and liver. It is strange that reference 4 is used here, which is the paper from Rebecca Siegel about Cancer Statistics.

The authors mention that “individuals often do not show symptoms during the early stages”. One should state that prostate cancer in its curable stages is asymptomatic. The symptoms that they might have with urination are due to BPH.

Reference 16 is twice used. In fact this is a paper on histopathology and not on what the authors mention.

Reference 7 is about Asian countries and reference 17 about active surveillance. This is incorrect referencing.

The authors mention that “there is support for providing routine PSA screening….”, reference 25 and 26 are from the United States (USPSTF and ACS). It would be nice to add the European view on this that has been presented through reference 125: EAU position and recommendations for 2021.

Finally in the background paragraph, there is mention of a Gleason system which ranges from 2 to 10. This is not in use any longer.

-          Discussion:

The demographical areas that show higher incidence: have the authors looked into the importance of sunshine (Vit D), next to the exposure to radiation?

Reference 126 is used to talk about issues related to over-diagnosis. There are obviously many more references than this one. The paper by Vickers et al, is advocating to either do an organized screening or apply screening only to symptomatic men, an issue that has been heavily criticized.

Reference 138 talks about decision making process although it is about the PROBASE trial in Germany who investigates whether screening should be started at 45 or 50.

Spelling error on line 376: prostate instead of prostrate

Line 381 “and resent study by Ma et al (143) indication”… should be: and a recent study by Ma et al (143) indicated…

The authors mentioned that “there is a need for education of patients and health workers”. In my view it is more important to have education of the healthy population that are not patients yet.

-          Conclusion:

The authors mention that “uptake for prostate cancer screening can be encouraged and improved by education”. This is correct, but the European Commission and the European Council want to stop opportunistic screening and replace it by organized screening. It would be nice to refer to the Swedish “organized PSA-testing” were despite information and education, in this high health literacy country makes that only 40 to 45% of men indeed have their PSA sampled. Look for Ola Bratt et al.

References:

Reference 23: the senior author is not M. Wirth, please check

Author Response

Reviewer 3

Response

What is the meaning of: “less of half of men reported for PSA screening”?

Edited- Less than half of men reported for PSA screening nonetheless there is a year-to-year increase in PSA and DRE screening

ABSTRACT

The uptake of prostate cancer screening needs to improve”: this is not what is the first aim of the European Commission today: they want to have organized screening programs, that indeed will be successful only when the health literacy among the male population has been upscaled.

The authors also mentioned that “there is the need to develop a comprehensive prostate cancer awareness program to assist in improving screening uptake and providing safe quality care”.  From experience in other countries awareness campaigns do not change very much to the low screening uptake, for instance in Sweden, the organized PSA-testing with an informed invitation letter for PSA testing resulted in no more than 40-45% of men finally being tested. This is pretty low compared to for instance the uptake in breast cancer screening.

In light of your relevant points, we have edited the conclusion of the abstract.

 Background:

The authors mention that “prostate cancer cells cause metastasis most commonly in bones, kidney, ureter and bladder”. For me, bone is the most affected, followed by lymph nodes, lung and liver. It is strange that reference 4 is used here, which is the paper from Rebecca Siegel about Cancer Statistics.

Thank you for the correction and the sentenced has be rearranged to “Affecting bone followed by lymph nodes, lung and liver. A suitable reference by “Chehal A. Duodenal Metastasis from Colorectal Cancer: A Case Report. Saudi J Pathol Microbiol. 2023;8(8):210-5. https://doi.org/10.36348/sjpm.2023.v08i08.004” has been identified for the corrections made.

The authors mention that “individuals often do not show symptoms during the early stages”. One should state that prostate cancer in its curable stages is asymptomatic. The symptoms that they might have with urination are due to BPH.

Thank you for your observation and comment, this was a mis phrasing where last resort should have been last step. This has now been edited to “which in its curable stages is asymptomatic.”

Reference 16 is twice used. In fact, this is a paper on histopathology and not on what the authors mention.

We appreciate your correction and reference has been duly changed to “Sakellakis M, Flores LJ, Ramachandran S. Patterns of indolence in prostate cancer. Experimental and Therapeutic Medicine. 2022,1;23(5):1-0. https://doi.org/10.3892/etm.2022.11278”

Reference 7 is about Asian countries and reference 17 about active surveillance. This is incorrect referencing.

Thank you for the correction and the references has been changed

7.) Huang J, Chan EO, Liu X, Lok V, Ngai CH, Zhang L, Xu W, Zheng ZJ, Chiu PK, Vasdev N, Enikeev D. Global Trends of Prostate Cancer by Age, and Their Associations with Gross Domestic Product (GDP), Human Development Index (HDI), Smoking, and Alcohol Drinking. Clin Genitourin Cancer. 2023. https://doi.org/10.1016/j.clgc.2023.02.003

And

17.) Chan JS, Lee YH, Hui JM, Liu K, Dee EC, Ng K, Liu T, Tse G, Ng CF. Long-term prognostic impact of cardiovascular comorbidities in patients with prostate cancer receiving androgen deprivation therapy: A population-based competing risk analysis. Int J Cancer. 2023;153(4):756‐764. doi:10.1002/ijc. 3455

The authors mention that “there is support for providing routine PSA screening….”, reference 25 and 26 are from the United States (USPSTF and ACS). It would be nice to add the European view on this that has been presented through reference 125: EAU position and recommendations for 2021.

We have included “The European Association of Urology (EAU) guidelines do not recommend widespread mass screening for prostate cancer but do strongly recommend early detection in men with positive health literacy.

Finally in the background paragraph, there is mention of a Gleason system which ranges from 2 to 10. This is not in use any longer.

Thank you for the correction, we have excluded the sentence as suggested.

Reference 126 is used to talk about issues related to over-diagnosis. There are obviously many more references than this one. The paper by Vickers et al, is advocating to either do an organized screening or apply screening only to symptomatic men, an issue that has been heavily criticized.

Reference 126 has been duly changed as suggested to “Van Poppel, H., Albreht, T., Basu, P. et al. Serum PSA-based early detection of prostate cancer in Europe and globally: past, present and future. Nat Rev Urol 19, 562–572 (2022). https://doi.org/10.1038/s41585-022-00638-6”

-          Discussion:

Reference 138 talks about decision making process although it is about the PROBASE trial in Germany who investigates whether screening should be started at 45 or 50.

We appreciate your correction and reference has been duly changed to “Cincidda, C.; Pizzoli, S.F.M.; Ongaro, G.; Oliveri, S.; Pravettoni, G. Caregiving and Shared Decision Making in Breast and Prostate Cancer Patients: A Systematic Review. Curr. Oncol. 2023, 30, 803-823. https://doi.org/10.3390/curroncol30010061”

Spelling error on line 376: prostate instead of prostrate

Spelling error has been corrected as suggested.

Line 381 “and resent study by Ma et al (143) indication”. should be: and a recent study by Ma et al (143) indicated…

Spelling error has been corrected as suggested.

The authors mentioned that “there is a need for education of patients and health workers”. In my view it is more important to have education of the healthy population that are not patients yet.

We have revised in light of your comment

Conclusion

The authors mention that “uptake for prostate cancer screening can be encouraged and improved by education”. This is correct, but the European Commission and the European Council want to stop opportunistic screening and replace it by organized screening. It would be nice to refer to the Swedish “organized PSA-testing” were despite information and education, in this high health literacy country makes that only 40 to 45% of men indeed have their PSA sampled. Look for Ola Bratt et al.

Conclusion edited and now acknowledges international differences among countries.

Reference 23: the senior author is not M. Wirth, please check.

Reference edited as final author name was missing.

Round 2

Reviewer 3 Report

none